# Identification of *ETV6-RUNX1*-like and *DUX4*-rearranged subtypes in paediatric B-cell precursor acute lymphoblastic leukaemia

Henrik Lilljebjörn[1], Rasmus Henningsson[2], Axel Hyrenius-Wittsten[1], Linda Olsson[1], Christina Orsmark-Pietras[1], Sofia von Palffy[1], Maria Askmyr[1], Marianne Rissler[1], Martin Schrappe[3], Gunnar Cario[3], Anders Castor[4], Cornelis J.H. Pronk[4], Mikael Behrendtz[5], Felix Mitelman[1], Bertil Johansson[1,6], Kajsa Paulsson[1], Anna K. Andersson[1], Magnus Fontes[2] & Thoas Fioretos[1,6]

Fusion genes are potent driver mutations in cancer. In this study, we delineate the fusion gene landscape in a consecutive series of 195 paediatric B-cell precursor acute lymphoblastic leukaemia (BCP ALL). Using RNA sequencing, we find in-frame fusion genes in 127 (65%) cases, including 27 novel fusions. We describe a subtype characterized by recurrent *IGH-DUX4* or *ERG-DUX4* fusions, representing 4% of cases, leading to overexpression of *DUX4* and frequently co-occurring with intragenic *ERG* deletions. Furthermore, we identify a subtype characterized by an *ETV6-RUNX1*-like gene-expression profile and coexisting *ETV6* and *IKZF1* alterations. Thus, this study provides a detailed overview of fusion genes in paediatric BCP ALL and adds new pathogenetic insights, which may improve risk stratification and provide therapeutic options for this disease.

[1] Department of Laboratory Medicine, Division of Clinical Genetics, Lund University, Lund 22184, Sweden. [2] Centre for Mathematical Sciences, Lund University, Lund 22362, Sweden. [3] Department of Pediatrics, University Hospital Schleswig-Holstein, Kiel 24105, Germany. [4] Department of Pediatrics, Skåne University Hospital, Lund University, Lund 22185, Sweden. [5] Department of Pediatrics, Linköping University Hospital, Linköping 58185, Sweden. [6] Department of Clinical Genetics, University and Regional Laboratories Region Skåne, Lund 22185, Sweden. Correspondence and requests for materials should be addressed to H.L. (email: henrik.lilljebjorn@med.lu.se) or to T.F. (email: thoas.fioretos@med.lu.se).

Paediatric B-cell precursor acute lymphoblastic leukaemia (BCP ALL), the most common childhood malignancy, is stratified into prognostically relevant genetic subgroups based on the presence of certain gene fusions and aneuploidies[1]. However, 25% of cases do not have any characteristic genetic aberrations at diagnosis, and the underlying driver events are unknown. For these cases, here denoted as 'B-other', the identification of pathogenetic changes will not only increase our understanding of the leukemogenic process, but may also be important in a clinical context, because such alterations can be used for improved risk classification and for targeted treatment. Recent genome-wide studies have provided critical pathogenetic insights into paediatric BCP ALL, including the identification of a dismal prognosis for cases with *IKZF1* deletions[2–5] and for cases with a 'Ph-like'[4–8] gene-expression signature similar to that of Philadelphia (Ph)-positive ALL. In addition, the mutational landscapes of BCP ALL subtypes defined by *ETV6-RUNX1*, *TCF3-PBX1*, *TCF3-HLF*, high hyperdiploidy (51–67 chromosomes), hypodiploidy (<45 chromosomes) or *MLL* (also known as *KMT2A*) rearrangements have been delineated using high-resolution sequencing techniques[9–13]. These studies have almost exclusively been performed at the DNA level and no large-scale characterization of the gene-fusion landscape in paediatric BCP ALL has been reported to date. To gain a better understanding of the gene-fusion landscape of BCP ALL, we performed RNA sequencing (RNA-seq) in a population-based series of 195 paediatric (<18 years of age) BCP ALL cases. We report that gene fusions are present in 65% of BCP ALL, and identify several new fusions and two novel subtypes; one characterized by recurrent *IGH-DUX4* or *ERG-DUX4* fusions and one characterized by an *ETV6-RUNX1*-like gene-expression profile, and coexisting *ETV6* and *IKZF1* alterations.

## Results

**Identified subtypes enable classification of 98% of cases.** All 195 cases subjected to RNA-seq had previously been analysed by G-banding, fluorescent *in situ* hybridization (FISH) and molecular analyses for the detection of established genetic BCP ALL alterations as part of routine clinical diagnostics (Supplementary Fig. 1 and Supplementary Data 1). Using RNA-seq, we identified an in-frame fusion gene in 127/195 (65%) BCP ALL cases and out-of-frame fusions in 20/195 (10%) cases (Fig. 1 and Supplementary Data 2–4). Notably, of the 68 cases lacking an in-frame fusion gene, the majority (64/68, 94%) were high-hyperdiploid (*n* = 56), hypodiploid (*n* = 2), Ph-like (*n* = 3), harboured a dic(9;20) (*n* = 1) or belonged to novel subtypes further described below (*n* = 2) (Fig. 1e and Supplementary Data 3). One subgroup, comprising 16% of B-other cases (4% of the entire BCP ALL cohort), harboured rearrangements of the double homeobox 4 (*DUX4*) gene and overlapped with a previously described group of patients with a homogenous gene-expression profile and frequent *ERG* deletions[6,14,15]. In addition, a new subtype, harbouring co-existing rearrangements of *ETV6* and *IKZF1* and associated with *ETV6-RUNX1*-like gene-expression pattern (3% of the cohort; 14% of B-other cases), was identified. Taken altogether, 98% of the BCP ALL cases could be classified into distinct genetic subtypes with a known underlying driver mutation or, less commonly, with a rare in-frame gene fusion (Figs 1f, 2, Supplementary Data 3), providing new insights and pathogenetic markers in BCP ALL.

**DUX4-rearranged cases constitute a distinct BCP ALL subtype.** Recurrent *DUX4* rearrangements were identified in 8/195 (4%) BCP ALL cases and were confined to B-other cases (8/50 cases, 16%; Figs 1a–c and 2, Supplementary Data 3). The

rearrangements were either a fusion between *IGH* and *DUX4* (7/8 cases) or between *ERG* and *DUX4* (1 case). To confirm this and other findings within the B-other group, we performed RNA-seq of an independent validation cohort of 49 paediatric B-other cases that were negative for *BCR-ABL1*, *ETV6-RUNX1*, *TCF3-PBX1*, *MLL* rearrangements and high hyperdiploidy (Supplementary Data 5). This analysis revealed an additional 20 cases with *DUX4* rearrangements, resulting in a total of 26 cases with *IGH-DUX4* and 2 with *ERG-DUX4* across the 2 cohorts.

*DUX4* encodes a homeobox-containing protein and is located within a subtelomeric D4Z4 repeat region on 4q and 10q. It is present in 11–100 copies on each allele, and is epigenetically silenced in somatic tissues. Loss of epigenetic silencing through shortening of the D4Z4 repeats leads to the degradation of muscle cells, and causes facioscapulohumeral muscular dystrophy[16,17].

To confirm the *DUX4* rearrangements at the genomic level, we performed mate-pair whole-genome sequencing (MP-WGS) in all eight cases in the discovery cohort, enabling powerful mapping of structural genomic rearrangements (Supplementary Data 6). These analyses confirmed the *DUX4* rearrangements at the DNA level in all cases, and revealed that the *IGH-DUX4* fusions resulted from insertions of a partial copy of *DUX4* into the *IGH* locus, including between 90–1,200 bp upstream of *DUX4* and between 939 and 1,272 bp of coding sequence from *DUX4* (Fig. 3 and Supplementary Fig. 2). Similarly, *ERG-DUX4* in case 75 was the result of an insertion of a partial copy of *DUX4* into intron 3 of *ERG*, containing 936 bp of coding sequence of *DUX4* (Fig. 3j). Neither *IGH-DUX4* nor *ERG-DUX4* would give rise to a chimeric protein; instead, the rearrangements and expression pattern suggest that the relocation of *DUX4* induces its expression from regulatory regions of the partner gene (Fig. 3 and Supplementary Fig. 3). The full-length DUX4 protein consists of 424 amino acids, but 7 of the 8 genomically characterized cases expressed truncated *DUX4* transcripts encoding between 312 and 420 amino acids (Fig. 3). Only case 47 expressed the full coding length of *DUX4*. All variants, however, retained both homeobox domains of DUX4, thus preserving its DNA-binding capacity.

All *DUX4*-rearranged cases displayed a distinct overexpression of *DUX4* as determined by RNA-seq; in contrast, expression of this gene was significantly lower or absent in the other investigated 216 BCP ALL cases across the discovery and validation cohorts (Supplementary Fig. 3). Notably, all cases with *DUX4* rearrangements displayed a global gene-expression pattern matching that of a subgroup of BCP ALL cases previously reported to be associated with *ERG* deletions in 38–55% of cases (Supplementary Fig. 4)[6]. Conversely, all cases with this gene-expression profile had *DUX4* rearrangements and overexpression of *DUX4*, indicating that the *DUX4* rearrangement is the founder event for this group (Supplementary Fig. 4). We determined the frequency of *ERG* deletions in *DUX4*-rearranged cases by MP-WGS in the discovery cohort and indirectly by ascertaining truncated transcripts by RT–PCR[14] in the validation cohort. This revealed *ERG* deletions in 5/8 (63%) cases in the discovery cohort and in 10/20 (50%) cases in the validation cohort (Supplementary Fig. 5), supporting that the *DUX4*-rearranged subtype reported here is identical to the previously described subtype with a distinct gene-expression profile and frequent *ERG* deletions[6]. This group has consistently been associated with a favourable prognosis, both when defined by the distinct gene-expression profile[6], and when defined by the characteristic *ERG* deletions[14,15]. In the discovery cohort, we observed no relapses among the 8 *DUX4*-rearranged cases, while 4 of 20 cases (20%) experienced relapse in the validation cohort. With the identification of *DUX4* rearrangement as a new marker in BCP ALL, it will be interesting to ascertain its prognostic impact in

larger, uniformly treated, cohorts. To characterize further the mutational landscape of *DUX4*-rearranged BCP ALL, whole-exome sequencing (WES) was performed in five *DUX4*-rearranged cases with matched constitutional samples available (Supplementary Data 6). These were found to harbour between 4 and 10 non-silent exome mutations each (Supplementary Data 7).

The only recurrently mutated gene was the transcription factor *ZEB2*, with mutations in two cases (#75 and #174).

A borderline significance for cases with *DUX4* rearrangements being older than cases lacking such fusions (Mann–Whitney's two-sided test, $P = 0.051$; median age 6.5 versus 4 years) was seen in the discovery cohort. The median age at diagnosis of patients

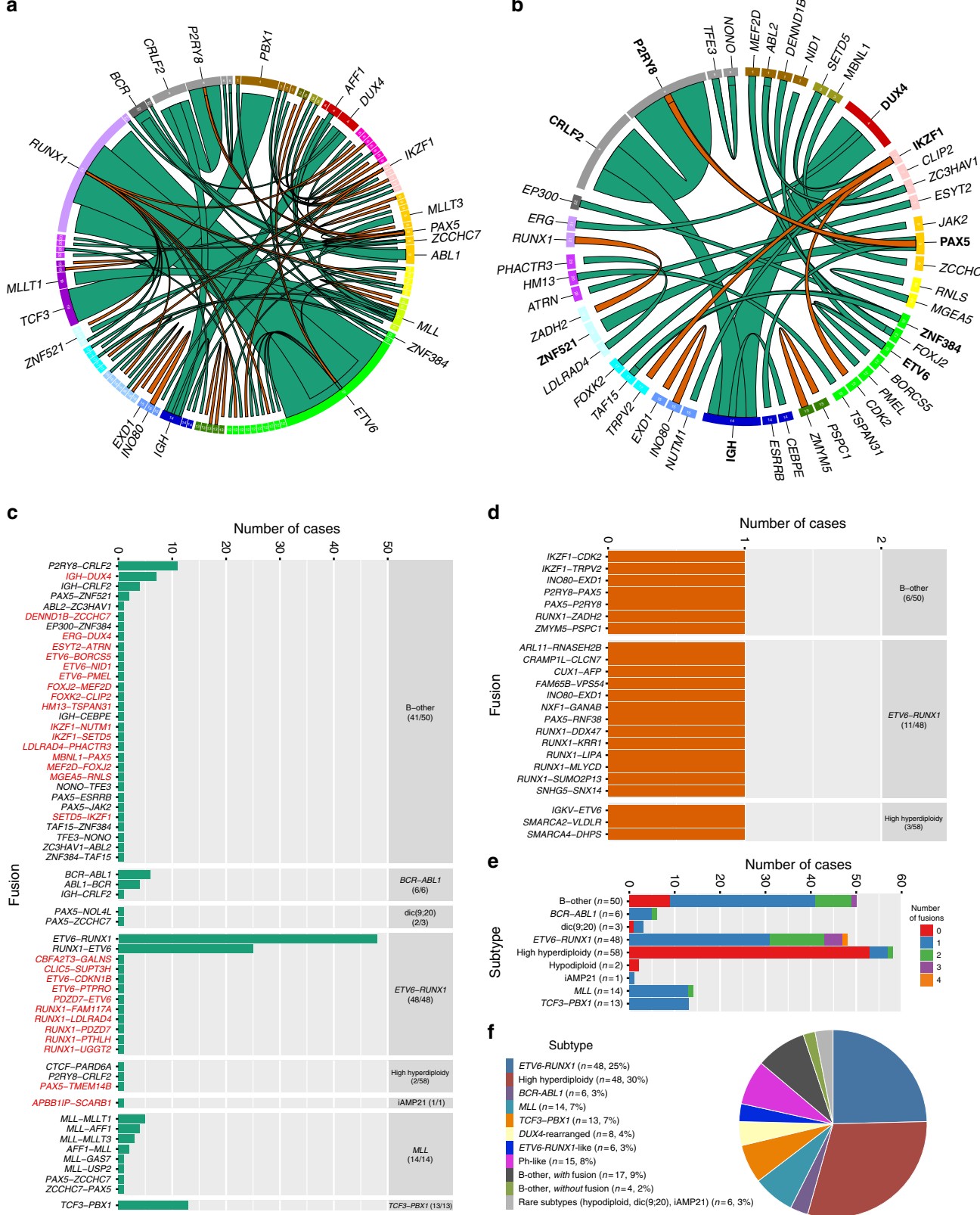

**Figure 1 | Overview of the gene fusions present in 195 paediatric BCP ALL cases in the discovery cohort.** (**a**) In-frame gene fusions (green) and out-of-frame gene fusions (orange) are illustrated using Circos[59]. Each ribbon has one end attached to the circle, indicating the 5′-partner gene of the fusion. The width of the ribbon is proportional to the number of detected fusions. Genes are arranged according to their genomic position (from chromosome 1–22 followed by X and Y) and chromosomes are marked in different colours. The gene symbol is denoted for genes involved in more than two unique fusions or in recurrent fusions. (**b**) In-frame gene fusions and out-of-frame gene fusions present in 50 B-other cases. The gene symbol for genes involved in more than two unique fusions or in recurrent fusions is indicated in bold. (**c**) The frequency of in-frame gene fusions by genetic subtype (indicated in the right column with the number of affected cases in parenthesis). Novel gene fusions are indicated in red ($n = 27$, reciprocal gene-fusion pairs counted as a single fusion) and previously described fusions are indicated in black ($n = 22$). (**d**) The frequency of out-of-frame gene fusions by genetic subtype (indicated in the right column with the number of affected cases in parenthesis). (**e**) Total number of gene fusions per case by genetic subtype (including both in-frame and out-of-frame fusions; reciprocal gene-fusion pairs counted as a single fusion). (**f**) Distribution of 195 BCP ALL cases within genetic subtypes defined by gene-expression profile and gene fusions detected by RNA-seq.

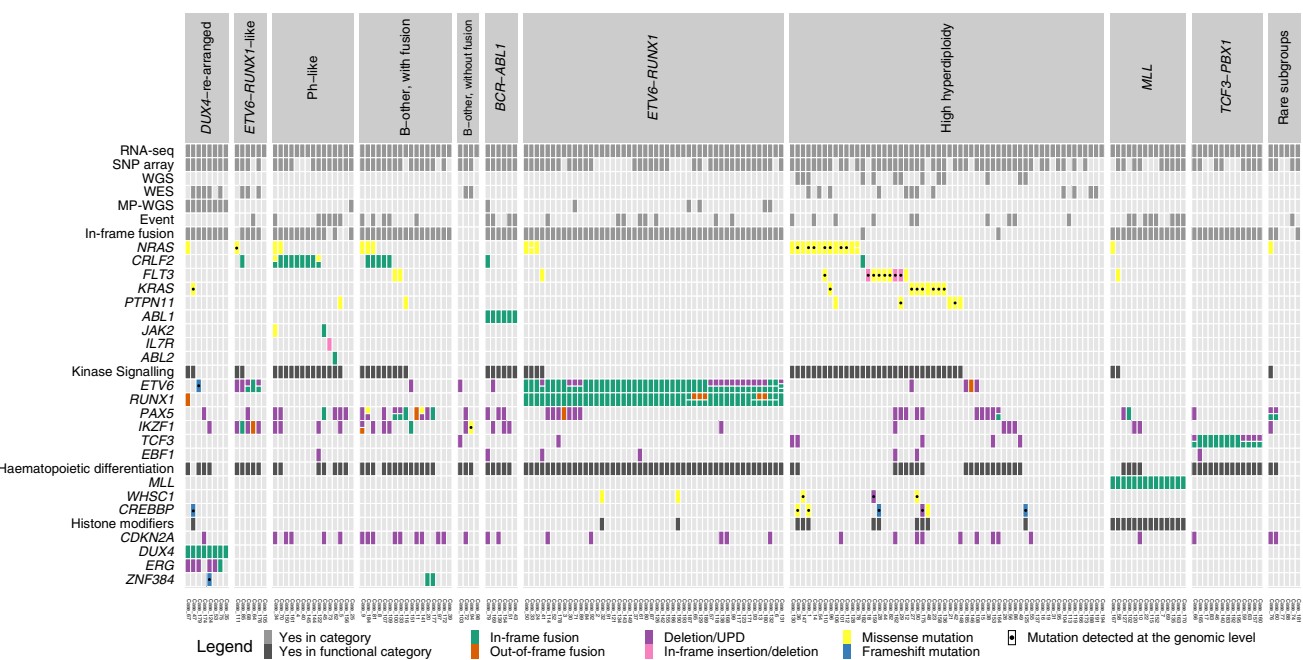

**Figure 2 | Genetic alterations present in 195 BCP ALL cases in the discovery cohort.** The cases are arranged according to genetic subtypes defined by gene-expression profile and gene fusions detected by RNA-seq, and were further characterized by SNP array, WGS, WES and MP-WGS. Genes recurrently altered in BCP ALL are arranged according to functional categories (kinase signalling, haematopoietic differentiation, histone modifiers and others). Events comprise induction failure and relapse.

with *DUX4*-rearranged ALL in the combined cohorts was 8.5 years (range 2–15 years). Considering the pronounced age peak at 3–5 years for childhood BCP ALL in general[18], this indicates that *DUX4*-rearrangments are associated with older age, although this needs to be confirmed in larger cohorts. Interestingly, an association with older age has previously been described for cases with *ERG* deletions[14,15].

The complexity of the genomic region where *DUX4* is located is most likely the reason that *DUX4* fusions have not been previously discovered in BCP ALL. Our standard RNA-seq bioinformatics pipeline could only detect the rearrangement in 7 of 28 cases, whereas a guided analysis that identified RNA-seq reads that linked any region within 2 kb of *DUX4* to the reads within the *IGH* locus identified the *IGH-DUX4* in an additional 19 cases (Supplementary Data 8). In the remaining two cases with *DUX4* overexpression, a fusion between *ERG* and *DUX4* was discovered by surveying the RNA-seq reads for regions similarly linked to the region surrounding *DUX4*. The aberrations were also not expected to be detectable on the chromosomal level by either G-banding or FISH, due to the small sizes of the insertions. In line with this, G-banding results from the eight *DUX4*-rearranged cases in the discovery cohort showed normal karyotypes in four cases and unspecific changes in two cases; in two cases, G-banding analyses had failed (Supplementary Data 1).

Taken altogether, RNA-seq followed by guided searches for *DUX4* chimeric transcripts is a reliable way to identify *DUX4* rearrangements, although both WES and MP-WGS allows the detection of the rearrangement at the genomic level (Supplementary Fig. 2).

***ETV6-RUNX1*-like gene expression in cases lacking the fusion.** Gene-expression profiling based on the RNA-seq data showed that 6/50 (12%) B-other cases in the discovery cohort clustered with the *ETV6-RUNX1*-positive cases, despite lacking molecular evidence of this fusion by FISH, RT–PCR and RNA-seq (Fig. 4a–c, and Supplementary Table 1). The gene-expression similarities were further supported by gene set enrichment analysis (GSEA)[19] (Supplementary Figs 6–8). These six cases were thus denoted '*ETV6-RUNX1*-like ALL'. Interestingly, RNA-seq together with single-nucleotide polymorphism (SNP) array profiling revealed that five of the six *ETV6-RUNX1*-like cases harboured co-existing *ETV6* and *IKZF1* aberrations (Figs 2, 5; Supplementary Data 3 and Supplementary Table 1).

Specifically, case 64 contained an in-frame fusion between *ETV6* (at 12p13) and *PMEL* (at 12q13) together with an out-of-frame fusion between *IKZF1* (at 7p12) and *CDK2* (at 12q13) that lacked functional domains from *IKZF1* (Fig. 5a

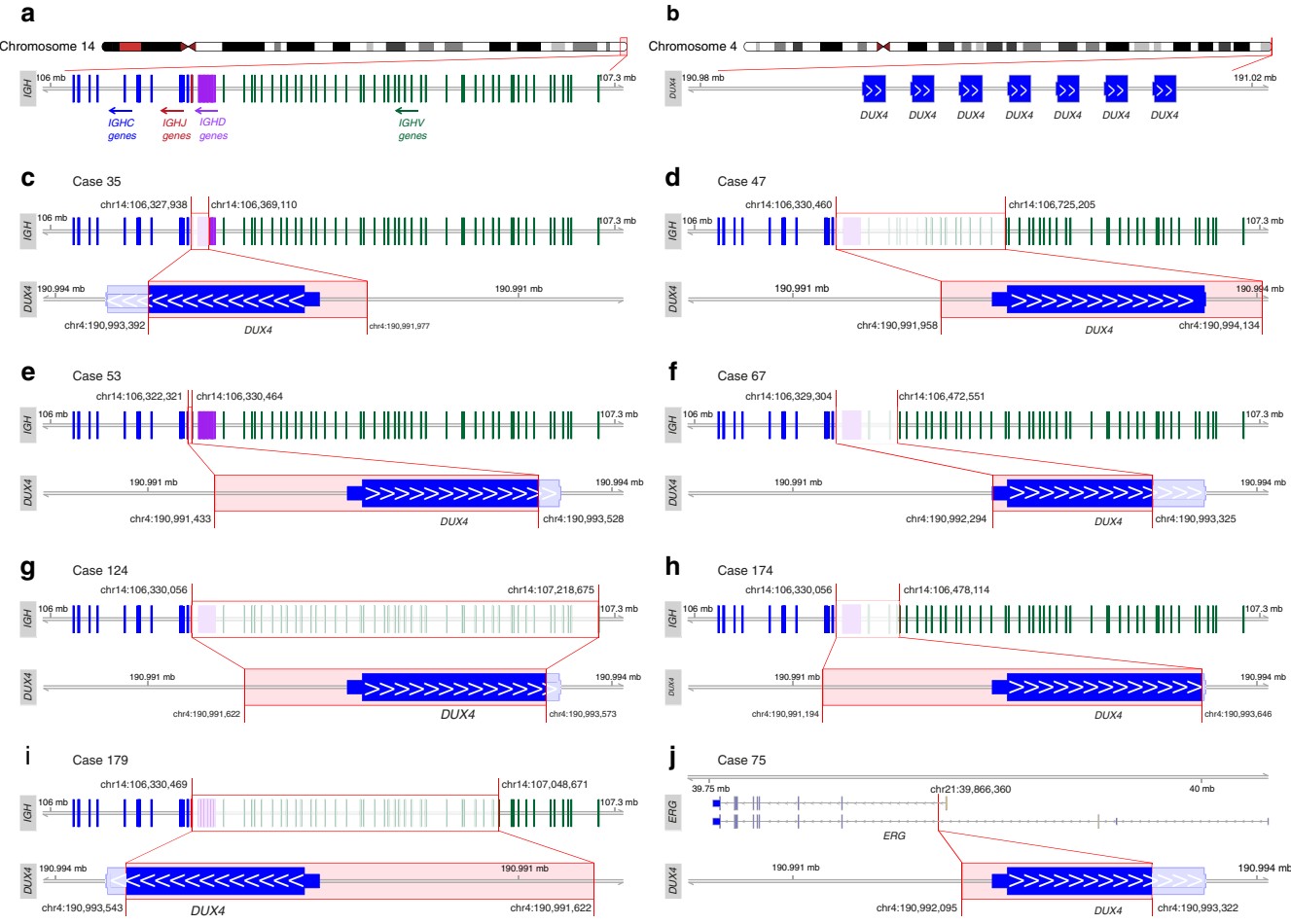

**Figure 3 | *DUX4* rearrangements in eight BCP ALL cases in the discovery cohort.** (**a**) Arrangement of immunoglobulin genes in the *IGH* locus. (**b**) Structure of the subtelomeric D4Z4 repeat region on 4q in the hg19 reference genome. This reference representation has seven repeats, each containing a *DUX4* gene. Healthy individuals have 11–100 repeats. (**c**–**i**) Structure of the *IGH-DUX4* rearrangements in (**c**) case 35, (**d**) case 47, (**e**) case 53, (**f**) case 67, (**g**) case 124, (**h**) case 174 and (**i**) case 179. (**j**) Structure of the *ERG-DUX4* rearrangement in case 75. All genomic coordinates are based on the human reference genome hg19. Because it is impossible to determine which *DUX4* repeat is involved in the rearrangement, the coordinates from the first *DUX4* repeat are represented in the figures.

and Supplementary Table 1). Case 68 contained an in-frame fusion between *ETV6* and *BORCS5* (12p13) caused by a small deletion in 12p13, together with a deletion spanning the first exons of *IKZF1* (Fig. 5b and Supplementary Table 1). Case 85 contained an intragenic *ETV6* deletion and a t(3;7)(p25;p12) giving rise to in-frame reciprocal *SETD5-IKZF1* and *IKZF1-SETD5* fusions (Fig. 5c and Supplementary Table 1). Case 111 had interstitial deletions on both 7p and 12p, resulting in whole-gene deletions of *IKZF1* and *ETV6* (Fig. 5d and Supplementary Table 1). Case 176 carried a deletion of the entire 7p, including *IKZF1*, together with an in-frame fusion between *ETV6* and *NID1* (at 1q42) and an interstitial deletion on 12p removing the second *ETV6* allele (Fig. 5e and Supplementary Table 1). Finally, one case (#105) had no lesions affecting *ETV6* or *IKZF1* as detected by RNA-seq (analysis by SNP array was precluded due to lack of DNA). Thus, in total, genetic lesions affecting both *ETV6* and *IKZF1* were identified in all five cases where both RNA-seq and SNP array profiling could be performed (Fig. 5 and Supplementary Table 1). Combined lesions of *ETV6* and *IKZF1* were otherwise exceedingly rare outside of this group (3/152 (2%) cases with available SNP array data; $P < 0.001$, Fisher's exact test; Fig. 2). To characterize further *ETV6-RUNX1*-like ALL, we performed WES on four *ETV6-RUNX1*-like cases with available matched constitutional samples (cases 68, 85, 111

and 176; Supplementary Data 6 and 7). These cases carried between 3 and 29 non-silent exome mutations (with an allele frequency above 10%), but no gene was recurrently mutated.

RNA-seq of the independent validation cohort identified four additional cases with *ETV6-RUNX1*-like gene-expression profiles. Three of these harboured out-of-frame *ETV6* fusions (with *CREBBP* at 16p13, *BCL2L14* at 12p13 and *MSH6* at 2p16); in the fourth case, no fusion was detected (Supplementary Data 5). Unfortunately, no DNA was available for SNP array analyses, precluding a complete evaluation of deletions affecting *ETV6* or *IKZF1* in these cases.

We conclude that alterations of *ETV6*, either by the generation of alternative gene fusions, or, more rarely, *ETV6* deletions, in combination with *IKZF1* lesions, represent an alternative mechanism to elicit the same transcriptional perturbation as seen in classical *ETV6-RUNX1* fusion-positive cases. Interestingly, both *IKZF1* and *RUNX1* encode transcription factors important for B-cell maturation[20,21], and it is tempting to speculate that loss of IKZF1 may substitute for the altered function of RUNX1 in the ETV6-RUNX1 fusion protein. In line with this, we note that *IKZF1* deletions are rare in the *ETV6-RUNX1*-positive cases (~3%) in this and other cohorts[22,23].

While the small number of *ETV6-RUNX1*-like cases prohibited meaningful survival analyses, only two relapses were recorded

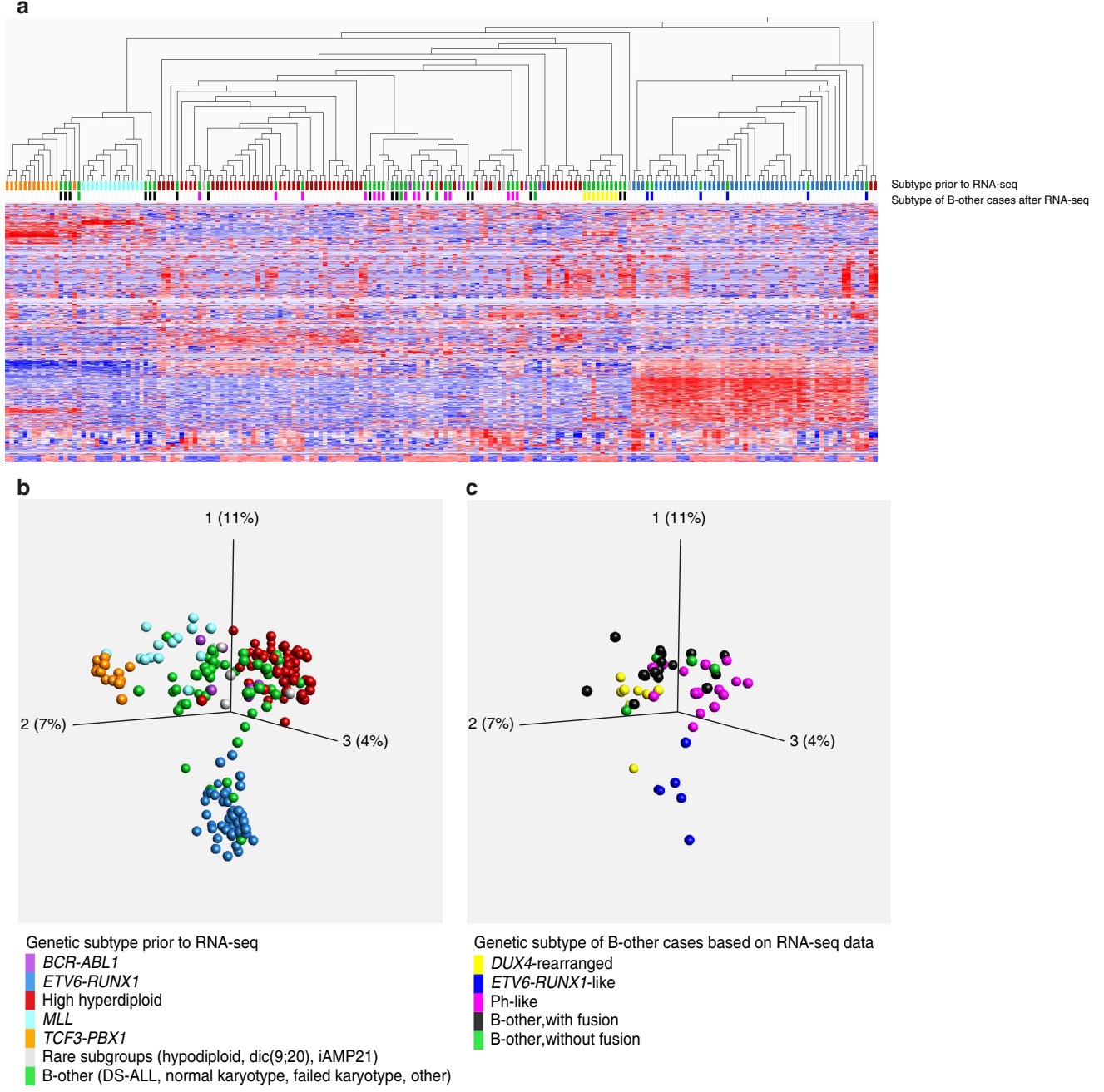

**Figure 4 | Hierarchical clustering and principal component analyses of RNA-seq gene-expression data.** A variance threshold was set at standard deviation 0.285, retaining 638 variables. The colour-coding of BCP ALL subtypes, used throughout the figure, is indicated in the bottom. (**a**) Unsupervised hierarchical clustering analysis of 195 BCP ALL cases. Coloured boxes below the dendogram indicate the subtype of each sample. The genetic subtype of B-other cases, based on the gene-expression and gene-fusion data, is indicated on the lower line. (**b**) Principal component analysis (PCA) of gene-expression data from all 195 BCP ALL cases. (**c**) PCA based on the data displayed in **b**, but only showing the 50 B-other cases colour-coded according to the genetic subtype based on the gene-expression and gene-fusion data. DS-ALL, Down's syndrome ALL; iAMP21, intrachromosomal amplification of chromosome 21.

among the ten *ETV6-RUNX1*-like cases in the combined discovery and validation cohort, indicating that the frequent *IKZF1* aberrations did not confer a dismal prognosis, as otherwise described for *IKZF1* deletions in BCP ALL[7,8]. However, further studies are warranted to evaluate the clinical impact of *IKZF1* deletions in *ETV6-RUNX1*-like BCP ALL.

**In-frame gene fusions are present in most B-other cases.** An in-frame fusion gene could be detected in 41/50 B-other cases (82%) in the population-based discovery cohort (Supplementary

Data 3). The B-other cases could be subdivided into five non-overlapping groups: those with Ph-like ($n = 15$; Supplementary Fig. 9a) or *ETV6-RUNX1*-like ($n = 6$) gene-expression profiles, those with *DUX4* rearrangements ($n = 8$), and remaining cases with ($n = 17$) or without ($n = 4$) in-frame gene fusions ('B-other, with fusion' and 'B-other, without fusion', respectively, Fig. 2).

In agreement with previous descriptions of Ph-like BCP ALL, most cases (11/15, 73%) harboured gene fusions that deregulate the cytokine receptor CRLF2 (*P2RY8-CRLF2*, $n = 6$; and *IGH-CRLF2*, $n = 3$) or activate therapeutically targetable kinases

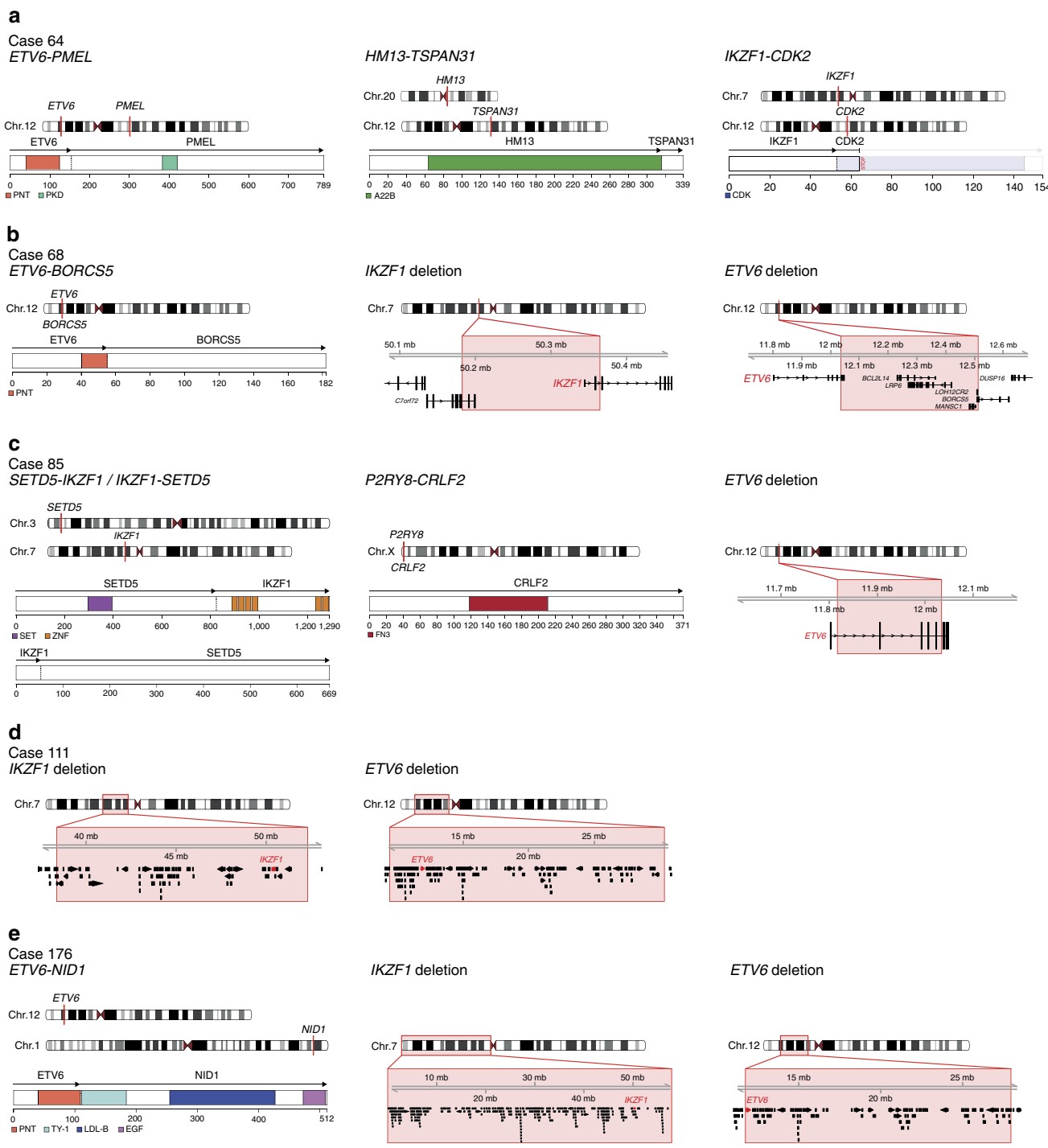

**Figure 5 | Overview of aberrations in BCP ALL cases with *ETV6-RUNX1*-like gene-expression pattern.** The fusion genes are illustrated with a schematic overview of the chromosomal position of the genes involved in the fusion (top) and the retained protein domains together (bottom). Genomic deletions affecting *ETV6* and *IKZF1* are depicted with a red box indicating the deletion both at the chromosomal level and at the gene level, with *ETV6* and *IKZF1* in red. Illustrated protein domains: PNT, pointed domain; PKD, polycystic kidney disease domain; A22B, Peptidase A22B domain; CDK, protein kinase domain; SET, SET domain; ZNF, zinc-finger domain; FN3, fibronectin type-III domain; TY-1, thyroglobulin type-1 domain; LDL-B, LDL-receptor class B repeats; and EGF, EGF-like domain. (**a**) Gene fusions present in case 64. No SNP array data were available for this case. *IKZF1-CDK2* is an out-of-frame fusion, with no functional domains from *CDK2* being included in the fusion protein. (**b**) Gene fusions and deletions present in case 68. The breakpoints of the *ETV6* deletion are within *ETV6* and *BORCS5*; likely representing the event that created the *ETV6-BORCS5* fusion gene. The breakpoints of the *IKZF1* deletion occur within the *C7orf72* and *IKZF1* genes, but RNA-seq data did not indicate the presence of a fusion transcript of these genes. (**c**) Gene fusions and deletions present in case 85. The *P2RY8-CRLF2* fusion does not contain any coding features from *P2RY8* but leads to overexpression of the entire coding region of *CRLF2*. (**d**) Deletions present in case 111. No gene fusions were detected in this case. (**e**) Gene fusions and deletions present in case 176.

(*ZC3HAV1-ABL2* in #62 and *PAX5-JAK2* in #45) (refs 7,8; Supplementary Data 3). In addition, RNA-seq data revealed mutations in the JAK-STAT pathway genes in 2/15 cases, 13% (Fig. 2 and Supplementary Data 3)[7,8].

Among the 17 cases in the 'B-other, with fusion' group, 11 cases (65%) harboured in-frame gene fusions previously described in BCP ALL[24]: *P2RY8-CRLF2* ($n = 4$), *PAX5-ZNF521* ($n = 2$), *EP300-ZNF384* (ref. 25) ($n = 1$), *IGH-CEBPE* ($n = 1$), *IGH-CRLF2* ($n = 1$), *PAX5-ESRRB* ($n = 1$) and *TAF15-ZNF384* ($n = 1$); in addition, a *NONO-TFE3* fusion gene, until now only reported in renal cell carcinoma[26,27], was found in a single case (#172; Fig. 1c and Supplementary Data 3). These fusions are likely genetic driver events in BCP ALL leukemogenesis. The importance of the novel in-frame gene fusions in the remaining five cases remains to be determined, but it is noteworthy that three had fusions (*DENND1B-ZCCHC7*, *MEF2D-FOXJ2*, *IKZF1-NUTM1*) involving genes recurrently rearranged in BCP ALL, namely *ZCCHC7*, *MEF2D*, *IKZF1* and *NUTM1* (ref. 24).

A high frequency of B-other cases from the validation cohort (36/49, 73%) also expressed an in-frame gene fusion. Using the same criteria as in the discovery cohort, the B-other cases in the validation cohort could be subdivided into *DUX4*-rearranged BCP ALL ($n = 20$), 'B-other, with fusion' ($n = 14$), 'B-other, without fusion' ($n = 7$), Ph-like ($n = 4$; Supplementary Fig. 9b) or *ETV6-RUNX1*-like ($n = 4$; Supplementary Fig. 10). Within the 'B-other, with fusion' group, ten cases harboured in-frame fusions previously described in BCP ALL: *EP300-ZNF384* ($n = 3$), *PAX5-FOXP1* ($n = 2$), *P2RY8-CRLF2* ($n = 2$, one of these cases also harboured *PAX5-FOXP1*), *PAX5-DACH1* ($n = 1$), *PAX5-ETV6* ($n = 1$), *TCF3-HLF* ($n = 1$) and *TCF3-ZNF384* ($n = 1$). Three of the remaining cases had a novel in-frame *MEF2D-HNRNPUL1* gene fusion and one case expressed a novel *MED12-HOXA9*. Hence, the majority of cases in the 'B-other, with fusion' group express recurrent gene fusions. Further studies are required to establish if these can be further stratified into biologically and clinically meaningful subtypes. However, we note that cases with fusions affecting each of the genes *ZNF384*, and *MEF2D* formed distinct but separate expression clusters by unsupervised hierarchical clustering, thus outlining possible subtypes characterized by similar gene fusions (Supplementary Figs 4, 10).

**Gene fusions in established genetic subgroups.** Most of the established genetic BCP ALL subgroups are based on recurrent gene fusions such as *BCR-ABL1*, *ETV6-RUNX1*, *TCF3-PBX1*, and *MLL* fusions. In the discovery cohort, the presence of these fusions had been ascertained by routine diagnostic analyses. By RNA-seq we could confirm these known gene fusions or their reciprocal variants in 77/81 (95%) cases (Supplementary Fig. 11). This implies that the RNA-seq analysis provided a relatively complete overview of the entire fusion-gene landscape, including also the novel identified fusions. The four instances of known gene fusions that could not be confirmed by RNA-seq were presumably caused by low expression of the fusion or rearrangements too complex for the analysis pipeline to elucidate.

High-hyperdiploid cases showed a notable lack of fusion genes, with only 2/58 cases in the discovery cohort harbouring in-frame fusion genes (3%, $P < 0.001$, Fisher's exact test; an additional three cases carried out-of-frame fusions), in accordance with our recent findings from WGS[11] (Figs 1c,d and 2). It was also uncommon for cases with *BCR-ABL1* ($n = 6$), *TCF3-PBX1* ($n = 13$) and *MLL* fusions ($n = 14$) to have additional in-frame or out-of-frame gene fusions, the only examples being the in-frame fusions *IGH-CRLF2* (in case 79 with *BCR-ABL1*) and *ZCCHC7-PAX5* (in case 102 with *MLL-GAS7*; Fig. 1c and Supplementary Data 3). In contrast, among the *ETV6-RUNX1*-positive cases, 6/48 cases (13%)

harboured in-frame fusions besides *ETV6-RUNX1* and its reciprocal variant, and 11/48 cases (23%) had out-of-frame fusions (Fig. 1c–e); the most commonly affected genes were *ETV6* ($n = 3$) and *RUNX1* ($n = 10$). Two of the *ETV6* fusions (with *CDKN1B* at 12p13 in #6 and *PTPRO* at 12p12 in #131; Supplementary Data 9) were formed by deletions affecting the *ETV6* allele not taking part in in the *ETV6-RUNX1* fusion. All ten *RUNX1* fusions had *RUNX1* as the 5'-partner gene and occurred in *ETV6-RUNX1*-positive cases lacking the reciprocal *RUNX1-ETV6* transcript (Supplementary Fig. 11), suggesting that they arose together with the *ETV6-RUNX1* fusion through a three-way translocation.

To characterize further the *RUNX1* fusions at the genomic level, 5/10 *ETV6-RUNX1*-positive cases containing additional *RUNX1* fusions were analysed by MP-WGS. These analyses confirmed the *RUNX1* fusions at the DNA level (Supplementary Data 6, 9 and 10) and revealed that the genomic breakpoints were in close proximity to the *RUNX1* breakpoints in the *ETV6-RUNX1* fusion, consistent with the presence of complex translocations. Such complex translocations have previously been detected in the *ETV6-RUNX1*-positive cases by FISH and targeted sequencing[28,29]. Only one *RUNX1* fusion contained an undisrupted active domain from the partner gene; thus, the fusions typically resulted in disruption of the 3'-partner gene (Supplementary Fig. 12).

**Fusion-gene network analysis.** To ascertain the pattern of gene fusions in BCP ALL, we performed a fusion-gene network analysis[30] of the 58 unique in-frame gene fusions identified across the discovery and validation cohorts (Supplementary Fig. 13a). This analysis revealed that 15 genes (*BCR*, *CRLF2*, *DUX4*, *ETV6*, *IGH*, *IKZF1*, *JAK2*, *LDLRAD4*, *MEF2D*, *MLL*, *PAX5*, *RUNX1*, *TCF3*, *ZCCHC7* and *ZNF384*) were recurrently involved in chimeras (Supplementary Fig. 13a). A comparison with literature data[24] highlighted that the high frequencies of fusions involving *RUNX1*, *DUX4*, *IKZF1* and *LDLRAD4* were novel findings (Supplementary Fig. 13a,b). The *RUNX1* fusions were typically found in *ETV6-RUNX1*-positive cases, most likely arising through complex translocations as described above, and the *DUX4* fusions were identified in the novel BCP ALL subgroup described in this study.

*IKZF1*, encoding IKAROS, is known to be perturbed by deletions (15% of BCP ALL cases) and occasionally sequence mutations (2–6%; refs 5,31,32), but has previously never been described to fuse with other genes in BCP ALL. In the discovery cohort, the two in-frame fusions *SETD5-IKZF1* (#85 with *ETV6-RUNX1*-like gene expression) and *IKZF1-NUTM1* (#151, "B-other, with fusion") retained functional domains from *IKZF1* (Supplementary Fig. 12e,f). *IKZF1-NUTM1* also contained essentially the entire coding region of *NUTM1*, akin to other *NUTM1* fusions in midline carcinoma[33] and in *MLL*-negative infant ALL[13]. The two out-of-frame fusions (*IKZF1-CDK2*, #64; and *IKZF1-TRPV2*, #9) contained no functional domains from *IKZF1* and, hence likely abolished the function of IKAROS. Thus, *IKZF1* fusions represent a novel mechanism for disrupting *IKZF1* in BCP ALL.

Aberrations in *LDLRAD4*, encoding a negative regulator of transforming growth factor-β signalling, have previously not been described in leukaemia. We identified two in-frame fusions involving this gene: *LDLRAD4-PHACTR3* (#70 with Ph-like gene expression) and *RUNX1-LDLRAD4* (#187 with *ETV6-RUNX1*). Both fusions retained the LDL-receptor class-A domain in the N-terminal region of the LDLRAD4 protein, whereas the SMAD interaction motif required for the regulation of transforming growth factor-β signalling was only retained in *RUNX1-LDLRAD4* (Supplementary Fig. 12a,d).

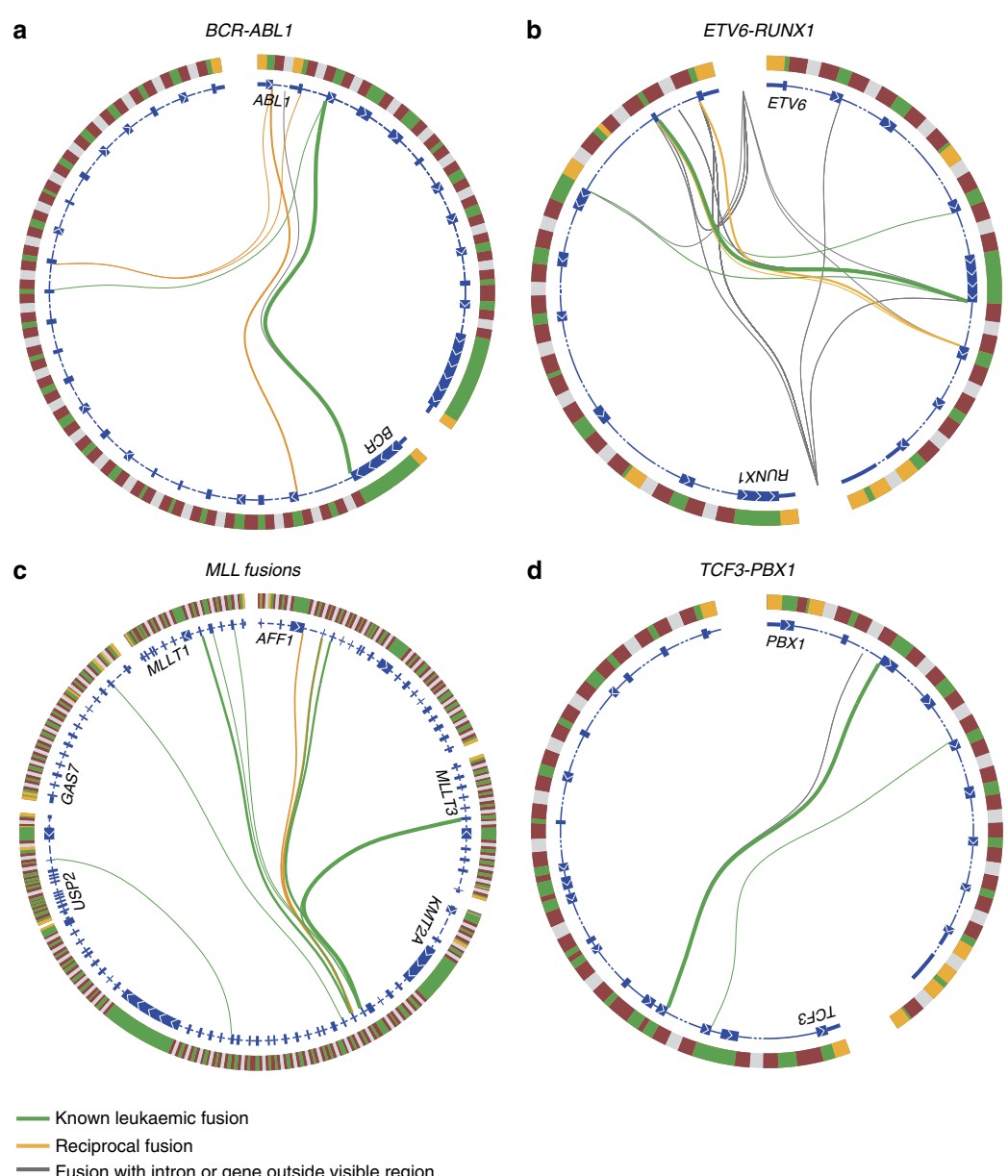

**Figure 6 | Splice patterns over fusion breakpoints.** Illustration of all detected fusion breakpoints in BCP ALL cases with (**a**) *BCR-ABL1*, (**b**) *ETV6-RUNX1*, (**c**) *MLL* fusions, and (**d**) *TCF3-PBX1*. Genes are arranged clockwise by genomic position. The outer circle represents the genomic region encompassing the indicated genes. Yellow indicates untranslated regions, green indicates coding exons, and red and grey indicate intronic regions (the latter are not to scale). The inner circle represents one or two overlaid reference transcripts of the indicated gene. Coding exons are indicated by a thick line with white arrows indicating the direction of the gene, introns are indicated by a thin or dashed line and untranslated regions are indicated by a medium thick line. Connecting lines between transcripts illustrate fusion breakpoints detected by at least three (for *BCR-ABL1*, *ETV6-RUNX1* and *MLL* fusions) or ten reads (for *TCF3-PBX1*). Fusion breakpoints in individual BCP ALL cases are depicted in Supplementary Figs 15–18.

**Intragenic splice variants and subtype classification.** Somatic intragenic deletions are frequent in BCP ALL and result in the expression of truncated transcripts predicted to encode internally deleted proteins[34]. To investigate if we could identify truncated transcripts associated with the most common intragenic deletions in BCP ALL (*CDNK2A*, *PAX5*, *ETV6* and *IKZF1*)[34], we developed a novel relative splice junction quantification algorithm. This algorithm identified five truncated transcript variants affecting *ETV6*, *PAX5* and *IKZF1*, with a total of 25 (13%) BCP ALL cases in the discovery cohort harbouring at least one truncated transcript (Supplementary Fig. 14). Focal deletions concordant with the truncating transcripts were present in 15/20 cases (75%) with available SNP array data. In five cases, the truncated transcript occurred without evidence of a focal deletion,

indicating either the presence of subclonal deletions below the detection level of the SNP array analysis or aberrant splicing caused by other mutational mechanisms.

Our detailed RNA-seq data also allowed analyses of splicing events occurring over the fusion breakpoints of the clinically important gene fusions *BCR-ABL1*, *ETV6-RUNX1*, *TCF3-PBX1* and *MLL* fusions, revealing a substantial heterogeneity in exon usage around the fusion breakpoints (Fig. 6a–d and Supplementary Figs 15–18); particularly for *ETV6-RUNX1* where the main variant joined exon 5 of *ETV6* with exon 2 of *RUNX1*, whereas alternative forms fused with exon 3 of *RUNX1* or a cryptic exon within intron 1 (Fig. 6b and Supplementary Fig. 16). These were either observed together as splice variants or as single forms in individual cases; the alternative variants did not affect

the runt domain of *RUNX1*. An alternative breakpoint joining exon 4 of *ETV6* with exon 2 of *RUNX1*, as has previously been described[35], was identified in 2/48 (4%) *ETV6-RUNX1*-positive cases (Supplementary Fig. 16).

Global gene-expression profiling by microarrays can discern between the genetic subtypes of BCP ALL, although with less than perfect accuracy[36–39]. We therefore constructed a classifier utilizing both gene-fusion and gene-expression data from RNA-seq. This classifier showed improved sensitivity (correctly classifying 180/195 cases, 92%) compared with a classifier based on the gene-expression data alone (correctly classifying 174/195 cases, 89%; Supplementary Fig. 19).

**Mutational analysis.** The mutational landscape of single-nucleotide variants in a larger number of genes has not been studied in an unselected series of BCP ALL. Because RNA-seq allows for the identification of expressed mutant alleles, we examined hotspot regions of 16 recurrently mutated genes in BCP ALL (representing 70% of all genes described to be mutated in more than 2 BCP ALL cases), ascertained in previous studies[9,11–13] or COSMIC[40] (Supplementary Data 11). This analysis revealed 56 mutations in 47 BCP ALLs, with genes in the RTK-RAS signalling pathway being the most commonly mutated: *NRAS* (23/195, 12%), *FLT3* (7/195, 4%), *PTPN11* (6/195, 3%) and *KRAS* (3/195, 2%; Supplementary Data 11). We also had genomic mutation data from 61 of the cases from WES ($n=22$), WGS ($n=12$), both WES and WGS ($n=1$), or Sanger sequencing ($n=26$; refs 11,41). We observed good concordance between hotspot mutations identified by RNA-seq and the genomic data, although some mutations in *KRAS* ($n=5$) and *FLT3* ($n=4$) observed at the DNA level escaped detection at the transcriptional level. The mutational spectra differed between subtypes, with *NRAS* and *KRAS* mutations being enriched in high-hyperdiploid cases[42], and *CRLF2*, *JAK2* and *IL7R* mutations in Ph-like ALL cases[7,8] (Fig. 2).

## Discussion

Gene fusions are strong driver mutations in neoplasia, and have provided fundamental insights into the disease mechanisms involved in tumourigenesis. In addition, they are increasingly used for diagnostic purposes, risk stratification and disease follow-up, and several chimeric proteins encoded by gene fusions serve as specific targets for treatment[30].

We here describe the gene-fusion landscape of paediatric BCP ALL, and show that the majority of cases (65%) express in-frame gene fusions, including most B-other cases (82%) previously described to lack specific genetic changes. The notable exception was high-hyperdiploid cases where only 3% of cases harboured an in-frame fusion gene. The low number of in-frame fusions in this group, however, highlights that the background level of gene-fusion generation in BCP ALL is low. Indeed, the median number of fusion genes per case in this study was 1 for all major subtypes ($>3$ cases) apart from high-hyperdiploid cases, showing that additional fusion genes are rarely needed for leukemogenesis. In *ETV6-RUNX1*-positive cases, however, additional fusions of unclear pathogenetic importance were present in 35% of cases. These typically involved *ETV6* or *RUNX1*. For the latter gene the fusions were generated through three-way translocations also creating the *ETV6-RUNX1* fusion.

We demonstrate, for the first time, that 16% of B-other cases (4% of BCP ALL) harboured rearrangements involving the *DUX4* gene. The frequency of such rearrangements differed between the discovery and validation cohorts; something that could possibly be explained by the higher mean age of the latter (7.1 versus 6.1 years). However, the true incidence of *DUX4* rearrangements in childhood BCP ALL needs to be further assessed in larger patient cohorts. The rearrangements resulted in fusions between *IGH* and *DUX4*, or less commonly, *ERG* and *DUX4*, causing aberrant *DUX4* expression. *DUX4* has previously only been reported to be rearranged in round-cell sarcomas, forming a recurrent *CIC-DUX4* fusion gene. That fusion, however, only includes a small C-terminal part of DUX4, not including the two homeobox domains[43], and is therefore likely to be functionally different from the fusions described here. Notably, all cases with *DUX4* rearrangements described herein displayed a gene-expression signature matching that of a subgroup of BCP ALL reported to be associated with frequent *ERG* deletions[6]. *DUX4* encodes a transcription factor normally expressed in germ cells that regulates the expression of genes involved in germline and early stem cell development[17,44]. Hence, it is tempting to speculate that the aberrant expression of DUX4 in the rearranged cases cause activation of transcriptional programmes that normally are expressed during early stem cell development. In contrast to the *ERG* deletions, *DUX4* rearrangements were present in all cases with the characteristic gene-expression pattern, implying that *DUX4* rearrangements constitute the founder event of this subtype and that *ERG* deletions are secondary cooperating events.

Global gene-expression profiling is a powerful tool to identify leukaemias with similar mutational backgrounds, as exemplified by the Ph-like subtype: such cases were initially identified as having gene-expression patterns similar to those of Ph-positive BCP ALL cases[4–6] and were only later identified as being characterized by genetic alterations that activate kinase or cytokine receptor signalling[7,8]. Within the B-other group we identified a second novel subtype, consisting of cases with a gene-expression profile similar to that of *ETV6-RUNX1*-positive cases but lacking this fusion gene. Instead, they harboured lesions affecting both *ETV6* and *IKZF1* in all cases with ascertainable data. We termed this subtype *ETV6-RUNX1*-like ALL. In contrast to cases with *ETV6-RUNX1*-like gene expression that have been reported in literature, but where cryptic *ETV6-RUNX1* were not excluded[45], we performed extensive genetic analyses to rule out a cryptic *ETV6-RUNX1* rearrangement. Thus, we propose that combined *ETV6* and *IKZF1* lesions together may activate similar transcriptional programmes as the ETV6-RUNX1 fusion protein.

The *DUX4*-rearranged and *ETV6-RUNX1*-like subtypes together with the well-established subgroup of Ph-like BCP ALL[4–8] accounted for 59 and 71% of B-other cases in the discovery and validation cohorts, respectively. Of the remaining B-other cases in the two cohorts, 74% expressed rare previously reported, or novel in-frame gene fusions, many of which contained genes with recurrent alterations in BCP ALL[24,40]. Again, these findings illustrate that paediatric BCP ALL, with the exception of the high-hyperdiploid, near-haploid and low-hypodiploid subgroups[11,12], is characterized by the presence of fusion genes. Because many gene fusions will be rare or even private, our study reinforces that RNA-seq may be a powerful tool for unbiased screening of fusion genes in a clinical setting with an unmatched power to detect novel but targetable gene fusions in BCP ALL[8,46].

In conclusion, this study provides a detailed view of the fusion gene landscape in paediatric BCP ALL, identifying several new gene fusions as well as distinct subgroups of BCP ALL. Apart from increasing our understanding of the pathogenesis of paediatric BCP ALL, this may help improve risk stratification and eventually increase the therapeutic options for this most common form of childhood malignancy.

## Methods

**Patients.** Between January 1992 and January 2013, 283 paediatric (<18 years) BCP ALL cases were analysed as part of clinical routine diagnostics at the Department of Clinical Genetics, University and Regional Laboratories Region

Skåne, Lund, Sweden. Of these, RNA or material suitable for RNA extraction from bone marrow ($n = 171$) or peripheral blood ($n = 24$) taken at diagnosis was available from 195 (69%) cases, comprising the discovery cohort. The vast majority was treated according to the Nordic Society of Paediatric Haematology and Oncology (NOPHO) ALL 1992, 2000 or 2008 protocols[47]. There were no significant differences in gender or age distribution between cases where RNA-seq could or could not be performed; however, cases analysed by RNA-seq had higher white blood cell counts (median $9.85 \times 10^9\,l^{-1}$, range $0.9-802 \times 10^9\,l^{-1}$ versus median $5.5 \times 10^9\,l^{-1}$, range $0.8-121 \times 10^9\,l^{-1}$; $P = 0.003$; two-sided Mann–Whitney's $U$-test). The validation cohort consisted of 49 paediatric BCP ALL cases treated according to the Berlin–Frankfurt–Münster (BFM) 2000 protocol[48]. All cases in the validation cohort were tested for BCR-ABL1, ETV6-RUNX1, TCF3-PBX1, MLL rearrangements, and high hyperdiploidy in accordance with the treatment protocol, and were found negative for these aberrations. Informed consent was obtained according to the Declaration of Helsinki and the study was approved by the Ethics Committee of Lund University.

**RNA sequencing.** The cDNA sequencing libraries were prepared from poly-A selected RNA using the Truseq RNA library preparation kit v2 (Illumina) according to the manufacturer's instructions, but with a modified RNA fragmentation step lowering the incubation time at 94 °C from 8 min to 10 s to allow for longer RNA fragments. The cDNA libraries were sequenced using a HiScanSQ (Illumina) or NextSeq 500 (Illumina).

**Gene-fusion detection.** Gene fusions were detected by combining three methods. Novel fusions were detected by Chimerascan[49] (0.4.5) and TopHat-Fusion-post[50] (2.0.7), followed by a custom filter strategy and validation by RT–PCR. Known fusion transcripts of BCR-ABL1, ETV6-RUNX1, TCF3-PBX1 and MLL fusions were detected by aligning all reads to a reference consisting of the known fusion transcripts and normal transcript variants of the genes, and counting reads uniquely aligned to the fusion transcripts. IGH-CRLF2 fusions were detected by identifying cases that had > 50 reads within a 65-kb region surrounding CRLF2 paired to a read within the IGH locus; these fusions were then validated using FISH. The following filter strategy was used for selection of validation candidates from Chimerascan and TopHat-Fusion-post results: all fusions reported by Chimerascan to be supported by ten or more reads over the fusion junction or > 50 total reads, all fusions reported by TopHat-Fusion-post to be supported by > 15 reads covering the fusion junction, and remaining interchromosomal fusions detected by Chimerascan that were also detected by TopHat-Fusion-post were included for validation, unless: (1) Chimerascan annotated the event as 'Read through'; (2) the affected exons had > 75% of reads mapped with a quality score below five; (3) reads supporting the same fusion were detected (by either TopHat or Chimerascan) in one of 20 sorted normal bone marrow samples; (4) the fusion indicated rearrangement within an IG or TCR locus or involved two HLA genes (the latter were presumed to represent normal constitutional HLA variants); (5) the fusion involved two non-coding genes; or (6) the constituent genes were located less than 10 kb apart.

In addition, remaining fusions detected by either Chimerascan or TopHat-Fusion-post were included if the fusion (1) was reciprocal to a fusion passing the above filters or a previously reported fusion in BCP ALL; or (2) contained one of the recurrently altered genes ETV6, RUNX1, MLL, PAX5 or IKZF1.

**Gene-expression analysis.** The raw unfiltered RNA-seq reads were aligned to human reference genome hg19 using TopHat 2.0.7, with the parameters --fusion-search and --bowtie1 to enable fusion detection. Gene-expression values were calculated as fragments per kilobase of transcript per million reads (fpkm) using Cufflinks 2.2.0 (ref. 51). Hierarchical clustering and principal component analyses were performed using Qlucore Omics Explorer (v3.1; Qlucore, Lund, Sweden). In brief, the data were normalized to a mean of 0 and a variance of 1. Hierarchical clustering of both samples and variables was performed using Euclidean distance and average linkage.

**Genomic sequencing analyses.** For 11 cases in the discovery cohort, whole-exome libraries were prepared from diagnostic and follow-up samples using the Nextera Rapid Capture Exome Kit (Illumina) according to the manufacturer's instructions. Paired $2 \times 151$ bp reads were produced from the exome libraries using a NextSeq 500 (Illumina). The reads were aligned to human reference genome hg19 using BWA 0.7.9a (ref. 52) and PCR duplicate reads were filtered out using SAMBLASTER[53]. Somatic variant calling was performed using Strelka[54]. For 15 cases in the discovery cohort, MP-WGS libraries were prepared using the Nextera Mate Pair Library Preparation Kit (Illumina). Paired $2 \times 76$ bp reads were produced from the mate-pair libraries using a NextSeq 500 (Illumina). The reads were aligned to human reference genome hg19 using BWA 0.7.9a (ref. 52) and PCR duplicate reads were filtered out using SAMBLASTER[53]. For 24 high-hyperdiploid cases in the discovery cohort, extensive characterization using WES ($n = 11$), WGS ($n = 12$) or both ($n = 1$) has been previously described[11].

**Identification of leukaemia-specific splice variants.** Splicing differences between samples were characterized by ascertaining the relative frequencies of splice junction usage across all observed splice donor and acceptor sites, from reads aligned by TopHat. All intragenic splice junctions that were supported by at least 10 reads in at least one sample and that involved at least one annotated exon were included. For each splice donor or acceptor site, alternative splicing was quantified by measuring the fraction of reads supporting each observed splice junction containing that site. If a splice acceptor or donor site was not covered by any reads within a sample, the corresponding variables were treated as missing values and reconstructed as the average value of samples that had data for the site. From these data, all splice variants in CDKN2A, PAX5, ETV6 and IKZF1 that were not present in a reference transcript and not detected in one of 20 normal bone marrow populations (sorted from four donors) were included in the analysis.

**Gene set enrichment analysis.** GSEA was performed on gene-expression data obtained from the RNA-seq analysis, using Qlucore Omics Explorer (v3.1). Signal-to-noise ratio was used as ranking metrics for analysing curated gene-ontology gene sets (C5) acquired from the Molecular Signatures Database (MSigDB). Gene sets with < 15 or > 500 genes were excluded. Enriched gene sets after 1,000 permutations at an false-discovery rate of < 0.25 and a nominal $P < 0.05$ were considered as significant.

**Support vector machine classification.** A classifier based on the gene-expression and gene-fusion data was created to categorize the samples into the subtypes BCR-ABL1, ETV6-RUNX1, high hyperdiploidy, MLL, TCF3-PBX1, and 'B-other + rare subgroups'. The subtypes were considered to be mutually exclusive. First, one-versus-all support vector machine classifiers[55] with linear kernels were created for all subtypes. They were based on the log2 transformed gene-expression data after variable selection by removal of variables with low variance across the samples. The threshold was set to a standard deviation of 0.29, resulting in 583/23,285 variables (2.5%) being used. Next, the classifiers were augmented by the detected gene fusions. If a gene fusion corresponding to one of the subgroups was found in a sample, it was classified as belonging to that subgroup regardless of the expression profile. These samples were treated as having $\pm \infty$ distance to the support vector machine classification hyperplane. Finally, a multiclass classifier was created from all the one-versus-all classifiers, by selecting the class that had the lowest signed distance between the sample and the classification hyperplane. The performance of the multiclass and all binary subgroup classifiers was evaluated by leave-one-out cross-validation.

**RNA-Seq mutation calling.** The raw unfiltered reads were aligned to human reference genome hg19 using STAR 2.4.0j (ref. 56). Putative mutations within hotspot regions of 16 genes were identified using VarScan 2.3.7 (ref. 57). The variants were annotated using Annovar[58] and known constitutional variants were excluded from the list.

**RT–PCR and Sanger sequencing.** For gene-fusion validation, primer3 was used to design primers for amplifying a region larger than 200 bp covering the fusion breakpoint. Reverse transcription was performed using M-MLV (Thermo Fischer Scientific) and PCR was performed using Platinum Taq (Thermo Fischer Scientific). The PCR products were purified using Exosap-it (Affymetrix) or Qiaquick gel extraction kit (Qiagen) and then Sanger sequenced by a commercial sequencing service provider. RT–PCR for detection of truncated ERG transcripts was performed using primers previously described[14]. Sanger sequencing of FLT3, NRAS, KRAS and PTPN11 in 26 high-hyperdiploid cases in the discovery cohort was performed using primers described in Supplementary Table 2. This data has been published previously[41].

**SNP array analysis.** SNP array analysis was performed on DNA extracted from bone marrow or peripheral blood at diagnosis for 156 BCP ALL cases. The analysis was performed using HumanOmni1-Quad and Human1M- Duo array systems (Illumina) with data analysis using Genomestudio 2011.1 (Illumina). The SNP array data has been previously published[3].

**Statistical methods.** Two-sided $P$ values were calculated using Fisher's exact test or Mann–Whitney's $U$-test. $P$-values of < 0.05 were considered statistically significant.

**Data availability.** RNA-seq and MP-WGS data have been deposited at the European Genome-phenome Archive (EGA), under the accession code EGAS00001001795. WES and WGS data are available for academic purposes by contacting the corresponding author, as the patient consent does not cover depositing data that can be used for large-scale determination of germline variants.

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

## Acknowledgements

This work was supported by the Swedish Cancer Society, the Swedish Childhood Cancer Foundation, the Swedish Research Council, the Knut and Alice Wallenberg Foundation, the Inga-Britt and Arne Lundberg Foundation, the Gunnar Nilsson Cancer Foundation,

the Medical Faculty of Lund University and Governmental Funding of Clinical Research within the National Health Service. We thank Birthe Fedders from the ALL-BFM laboratory and Andrea Biloglav from the Division of Clinical Genetics, Lund, for expert technical assistance.

## Author contributions

H.L. and T.F. conceived the project. H.L., A.H.-W., L.O., C.O.-P. M.A. and M.R. performed the experiments. H.L., R.H., A.H.-W., L.O., C.O.-P., S.v.P., F.M., B.J., K.P., A.K.A., M.F. and T.F. analysed the data. M.S., G.C., A.C. C.J.H.P. and M.B. provided samples and clinical data. H.L., K.P., A.K.A. and T.F. wrote the manuscript, which was reviewed and edited by the other co-authors.

## Additional information

**Competing financial interests:** The authors declare no competing financial interests.

