## [Peer Review File · Nature Communications]

Reviewer #3 (Remarks to the Author):

The revised manuscript has markedly improved in the quality of data presentation, clarity of message and importantly with the description and further characterization of the IGH-DUX and ERG-DUX fusions increased in novelty. All main reviewers comments have been addressed with only some minor points warranting further consideration.

1. Emphasis on comprehensive characterization

The authors should ensure that the novelty of this study is predominantly centered on the B-other subtypes

2. It is intriguing that the validation dataset showed such a strong enrichment of DUX4 rearrangements - the authors should comment on the selection criteria of the validation cohort and why such a strong difference in frequency of these events was observed. For example in the first cohort 8 cases were identified whereas in the validation 20/49 harbored these rearrangements.

3. The authors should provide more detail into the analysis and data derived from whole exome and whole genome studies. They should provide metrics (depth/reads etc) and importantly highlight what specific analysis each dataset was used for. For example did they authors identify DUX rearrangements by WES as they state on page 7 last sentence of paragraph 2. That would be very surprising.

4. Treatment outcome. Whilst the authors postulate that they included a validation cohort of more uniformly treated patients that is more representative of current outcomes the authors in line with clarification of point 2 - should state that the clinical significance of all these subtypes warrants further evaluation in larger, uniformly treated and representative cohorts.

5. Complex three way events have previously been reported in ETV6-RUNX1 ALL by whole genome sequencing studies focusing on these subtypes. The reviewers should acknowledge these.

Reviewer #4 (Remarks to the Author):

Lilljebjörn and colleagues performed RNA sequencing on 195 cases of BCP ALL to define the gene fusion landscape and found in-frame gene fusions in 125 (65%) cases, including 27 novel fusions. They identified two novel subtypes. The one is characterized by recurrent IGH-DUX4 or ERG-DUX4 fusions, representing 4% of cases. The other characterized by ETV6-RUNX1-like gene expression profile coexisting ETV6 and IKZF1 alterations with favorable outcome.

Their study is very nice. Their findings are novel, interesting and may appeal to a wide audience. Their methodology is appropriate and the paper is well organized. The authors have responded to the comments of the previous reviews sufficiently and the manuscript is significantly improved.

Major points:

I think that the important discovery of this study is the identification of two characteristic subtypes of B-others, namely B-others harboring DUX4-related fusions and ETV6-RUNX1-like BCP-ALL. However, I am feeling that it is over-interpretation to say "the gene fusion landscape". As the Reviewer #1 stated (89507 0 rebuttal), the sample size of 195 may be too small or not representative to define the gene fusion landscape of pediatric BCP-ALL. In fact, the authors stated other characteristic subtypes, such as clustered in R5 gene set and R3 gene set presented in Supplementary Figure 4. If investigate with more large size of the patients, these groups may constitute distinct subtypes in "B-others, with fusion". In addition, although the authors stated in Abstract, lines 9-11 "this study provides a comprehensive overview of gene fusions in pediatric BCP ALL and adds new pathogenetic insights, which should improve risk stratification and therapeutic options in this disease.", their evidences seem to be insufficient to discuss the significance of two subtypes in risk stratification.

Please reply to above questions of this reviewer. Otherwise, I recommend to modify the title and re-organize the manuscript in the form featuring the identification of two novel subtypes of B-others.

Minor points:

1) For this reviewer, it is unclear that why and how the validation cohort was selected. I could not find the description for the validation cohorts.

2) The sentences p3, lines 22-23, "Thus, the total frequency of cases that could be assigned to a genetic subtype or had an in-frame fusion gene was 98%." and p4, lines 6-9, "Taken together, 98% of the BCP ALL cases could be classified into distinct genetic subtypes with a known underlying driver mutation or, less commonly, with a rare in-frame gene fusion, providing new insights and markers of importance in BCP ALL." seem to be repeated description.

Response to Reviewers Comments and Questions

NCOMMS-16-04796-T “The gene fusion landscape of pediatric B-cell precursor acute lymphoblastic leukemia.”

We would like to thank the editor and the reviewers for the positive response on our revised manuscript and the thoughtful and constructive comments that we feel have significantly improved our study.

Please find our detailed response to each point raised by the reviewers outlined below. All corresponding changes in the manuscript Word-file are marked using the “tracking changes” function.

Reviewer #3 (Remarks to the Author):

The revised manuscript has markedly improved in the quality of data presentation, clarity of message and importantly with the description and further characterization of the IGH-DUX and ERG-DUX fusions increased in novelty. All main reviewers comments have been addressed with only some minor points warranting further consideration.

1. Emphasis on comprehensive characterization

The authors should ensure that the novelty of this study is predominantly centered on the B-other subtypes

Response: We agree, and to further underscore the importance of our novel findings we have changed the title to "Novel *ETV6-RUNX1*-like and *DUX4*-rearranged subtypes in pediatric B-cell precursor acute lymphoblastic leukemia". In addition, as requested by the Editor, we have added the following paragraph at the end of the Introduction to further highlight our results and conclusions (page 3, line 10):

*“We report that gene fusions are present in 65% of BCP ALL and identify several new fusions and two novel subtypes; one characterized by recurrent *IGH-DUX4* or *ERG-DUX4* fusions and one characterized by an *ETV6-RUNX1*-like gene expression profile and coexisting *ETV6* and *IKZF1* alterations.”*

*2. It is intriguing that the validation dataset showed such a strong enrichment of *DUX4* rearrangements - the authors should comment on the selection criteria of the validation cohort and why such a strong difference in frequency of these events was observed. For example in the first cohort 8 cases were identified whereas in the validation 20/49 harbored these rearrangements.*

Response: Unlike the discovery cohort, the validation cohort does not represent a population-based series. The discovery cohort represents a consecutive and population-based series of BCP-ALL referred to our Clinical Department for chromosome analysis and molecular studies as part of routine clinical diagnostic procedures. The validation cohort was selected from the German ALL study group BCP ALL. To match the B-other group, cases lacking *BCR-ABL1*, *ETV6-RUNX1*, *TCF3-PBX1*, *MLL* rearrangements, and high hyperdiploidy were selected, but there was also a strong focus on available material. Hence, we cannot rule out that there has been a selection bias in the validation cohort. Possibly indicating such a bias, the mean age for the 49 cases within the validation cohort was higher than for the 50 cases in the discovery cohort (mean age 7.1 vs 6.1 years). This could conceivably explain the higher

incidence of *DUX4*-rearrangements. This information has now been added to the discussion section (page 17, paragraph 3, line 2):

“We demonstrate, for the first time, that 16% of B-other cases (4% of BCP ALL) harbored rearrangements involving the *DUX4* gene. This frequency differed between the discovery and validation cohorts; something that could possibly be explained by the higher mean age of the latter (7.1 years vs. 6.1 years). However, the true incidence of *DUX4*-rearrangements in childhood BCP ALL needs to be further assessed in larger patient cohorts.

3. The authors should provide more detail into the analysis and data derived from whole exome and whole genome studies. They should provide metrics (depth/reads etc) and importantly highlight what specific analysis each dataset was used for. For example did they authors identify DUX rearrangements by WES as they state on page 7 last sentence of paragraph 2. That would be very surprising.

Response: We have added “Supplementary Data 6” that presents metrics for the whole exome and whole genome data as well as information on what analysis each dataset was used for. Regarding identifying *DUX4* rearrangements using WES data, we were also surprised to find that all rearrangements were clearly visible within the WES data. The reason the rearrangements were visible was that the breakpoints occurred close to, or even within, the first exon of *DUX4*. Apart from adding “Supplementary Data 6”, this has also been clarified by adding a reference to “Supplementary Figure 2” after the above sentence. “Supplementary Figure 2” shows the information provided by the WES, RNA-seq, and MP-WGS data with regards to the *DUX4* breakpoints.

4. Treatment outcome. Whilst the authors postulate that they included a validation cohort of more uniformly treated patients that is more representative of current outcomes the authors in line with clarification of point 2 - should state that the clinical significance of all these subtypes warrants further evaluation in larger, uniformly treated and representative cohorts.

Response: We agree that the clinical significance of the two novel subtypes needs to be addressed in larger more uniformly treated cohorts, in particular for the *ETV6-RUNX1*-like subtype and as pointed out by this reviewer, we have already added a statement that the clinical impact of this subtype needs to be determined in larger studies (last sentence page 10, first section). The overlap between the *DUX4*-rearranged subtype and the previously studied group with *ERG*-deletions and a uniform gene expression pattern, which has been shown to have a superior prognosis, also supports that this subtype is associated with a good prognosis. However, we have included a statement that its prognostic impact should be ascertained in larger studies of uniformly treated patients (page 6, line 12):

“This group has consistently been associated with a favorable prognosis, both when defined by the distinct gene expression profile⁶, and when defined by the characteristic *ERG* deletions^{15,16}. In the discovery cohort, we observed no relapses among the 8 *DUX4*-rearranged cases, while 4 of 20 cases (20%) experienced relapse in the validation cohort. With the identification of *DUX4* rearrangement as a new marker in BCP ALL, it will be interesting to ascertain its prognostic impact in larger, uniformly treated, cohorts.”

And with regards to the *ETV6-RUNX1*-like subtype on page 9, paragraph 4:

“While the small number of *ETV6-RUNX1*-like cases prohibited meaningful survival analyses, only two relapses were recorded among the ten *ETV6-RUNX1*-like cases in the combined discovery and validation cohort, indicating that the frequent *IKZF1* aberrations did not confer a dismal prognosis, as otherwise described for *IKZF1* deletions in BCP ALL^{7,8}. However, further studies are warranted to evaluate the clinical impact of *IKZF1* deletions in *ETV6-RUNX1*-like BCP ALL.”

5. Complex three way events have previously been reported in ETV6-RUNX1 ALL by whole genome sequencing studies focusing on these subtypes. The reviewers should acknowledge these.

Response: We cannot find any information on complex three-way translocations in the largest published whole genome sequencing study of *ETV6-RUNX1*-positive cases (by Papaemmanuil et al, ref 9 in the manuscript), presumably because this study focused on structural events occurring after the formation of *ETV6-RUNX1*. Other studies have, however, identified such translocations and this is now acknowledged in the manuscript (page 13, paragraph 2, line 6):

“Such complex translocations have previously been detected in *ETV6-RUNX1*-positive cases by FISH and targeted sequencing^{29,30,}”

Reviewer #4 (Remarks to the Author):

*Lilljebjörn and colleagues performed RNA sequencing on 195 cases of BCP ALL to define the gene fusion landscape and found in-frame gene fusions in 125 (65%) cases, including 27 novel fusions. They identified two novel subtypes. The one is characterized by recurrent *IGH-DUX4* or *ERG-DUX4* fusions, representing 4% of cases. The other characterized by *ETV6-RUNX1*-like gene expression profile coexisting *ETV6* and *IKZF1* alterations with favorable outcome.*

Their study is very nice. Their findings are novel, interesting and may appeal to a wide audience. Their methodology is appropriate and the paper is well organized. The authors have responded to the comments of the previous reviews sufficiently and the manuscript is significantly improved.

Major points:

*I think that the important discovery of this study is the identification of two characteristic subtypes of B-others, namely B-others harboring *DUX4*-related fusions and *ETV6-RUNX1*-like BCP-ALL. However, I am feeling that it is over-interpretation to say "the gene fusion landscape". As the Reviewer #1 stated (89507 0 rebuttal), the sample size of 195 may be too small or not representative to define the gene fusion landscape of pediatric BCP-ALL. In fact, the authors stated other characteristic subtypes, such as clustered in R5 gene set and R3 gene set presented in Supplementary Figure 4. If investigate with more large size of the patients, these groups may constitute distinct subtypes in "B-others, with fusion". In addition, although the authors stated in Abstract, lines 9-11 "this study provides a comprehensive overview of gene fusions in pediatric BCP ALL and adds new pathogenetic insights, which should improve risk stratification and therapeutic options in this disease.", their evidences seem to be insufficient to discuss the significance of two subtypes in risk stratification.*

Please reply to above questions of this reviewer. Otherwise, I recommend to modify the title and re-organize the manuscript in the form featuring the identification of two novel subtypes of B-others.

Response: We agree that the two novel subtypes are the most important findings of this study. To highlight this we have now changed the title of the manuscript to "Novel *ETV6-RUNX1*-like and *DUX4*-rearranged subtypes in pediatric B-cell precursor acute lymphoblastic leukemia". We have also, as requested by the editor, added a paragraph describing these findings at the end of the Introduction. The novel findings are also highlighted within the Results section where the two largest subsections are dedicated to the description of the novel subtypes.

However, we also believe that the data from the total series of almost 200 cases - the largest population-based BCP ALL series where systematic fusion gene detection has been performed to date - is powerful enough to warrant the description "gene fusion landscape". We note that this terminology is common in similar sized, and even smaller, studies published in high impact journals, such as those of melanoma (135 cases; Hodis et al, Cell, 2012), acute myeloid leukemia (200 cases; The Cancer Genome Atlas Network, NEJM, 2013), and esophageal squamous cell carcinoma (113 cases; Gao et al, Nat. Genet., 2014). More importantly, we do expect to find recurrent aberrations with a frequency >1% within the data, and can thereby - in our opinion - provide a detailed overview of fusion genes in BCP ALL.

We also agree that the wording "*which should improve risk stratification and therapeutic options in this disease*" in the abstract could be considered too optimistic, and we have therefore changed this to "*which may improve risk stratification and provide novel therapeutic options in this disease*". We believe that the wording in this last sentence of the abstract is appropriate since 1) we demonstrate that the *DUX4*-rearranged subtype overlaps with a group of cases that are characterized by *ERG* deletions and that previously have been described to have a good prognosis, and 2) cases with an *ETV6-RUNX1*-like gene expression profile are identified as genetically separate from Ph-like ALL cases. Hence, the identification of *ETV6-RUNX1*-like cases may improve risk stratification, at a minimum by identifying B-other cases that are not Ph-like, as these are associated with a dismal prognosis.

Minor points:

1) *For this reviewer, it is unclear that why and how the validation cohort was selected. I could not find the description for the validation cohorts.*

Response: To clarify why and how the validation cohort was selected, we have included the following paragraph in the results section of the manuscript (page 4, paragraph 2, line 4):

*"In order to confirm this and other findings within the B-other group, we performed RNA-seq of an independent validation cohort of 49 pediatric B-other cases that were negative for *BCR-ABL1*, *ETV6-RUNX1*, *TCF3-PBX1*, *MLL* rearrangements, and high hyperdiploidy (Supplementary Data 5)."*

2) *The sentences p3, lines 22-23, "Thus, the total frequency of cases that could be assigned to a genetic subtype or had an in-frame fusion gene was 98%." and p4, lines 6-9, "Taken together, 98% of the BCP ALL cases could be classified into distinct genetic subtypes with a known underlying driver mutation or, less commonly, with a rare in-frame gene fusion, providing new insights and markers of importance in BCP ALL." seem to be repeated description.*

Response: The first of these sentences has now been deleted to avoid repetition.